# Endoscopic-Assisted Keyhole Middle Cranial Fossa Approach for Small Vestibular Schwannomas

**DOI:** 10.3390/jcm11092324

**Published:** 2022-04-21

**Authors:** In Seok Moon, Ick Soo Choi, Seung Ho Shin, Seungjoon Yang, Youngrak Jung, Gina Na

**Affiliations:** 1Department of Otorhinolaryngology, Yonsei University College of Medicine, Seoul 03722, Korea; ismoonmd@yuhs.ac (I.S.M.); shins@yuhs.ac (S.H.S.); tmdwns2482@yuhs.ac (S.Y.); dudfkr01@yuhs.ac (Y.J.); 2Department of Otorhinolaryngology, Inje University College of Medicine, Ilsan Paik Hospital, Goyang 10380, Korea; goodnose@paik.ac.kr

**Keywords:** vestibular schwannoma, minimally invasive, middle cranial fossa, endoscope

## Abstract

The classical middle cranial fossa approach (MCFA) for vestibular schwannoma (VS) removal often requires a large incision and craniotomy, excessive temporal lobe manipulation, and a longer recovery. We describe a keyhole MCFA (KMCFA) with endoscopic assistance that allows for adequate access with minimal temporal lobe manipulation, resulting in a fast recovery and an invisible scar. Eight sides of four cadaveric heads were dissected through the endoscopic-assisted KMCFA to access the internal auditory canal (IAC). Furthermore, five patients with intracanalicular VS underwent tumor removal with the endoscopic-assisted KMCFA. During the endoscopic-assisted KMCFA with fine instruments, a 3-cm supra-auricular incision and a 2-cm diameter keyhole craniotomy achieved exposure of the entire length of the IAC in all cadaveric dissections without unintended violation of the cochlea, semicircular canal, and facial nerve. The gross tumor was totally removed in five patients with no major postoperative complications. The surgical time was reduced, the hearing outcomes were similar to those of the classical MCFA, and the scar was invisible 1 month after the surgery. The endoscopic-assisted KMCFA permits intracanalicular VS removal in a safe, efficient, and cosmetic way. For small intracanalicular VSs, this approach can replace the classical MCFA when indicated.

## 1. Introduction

Since its technical refinement by William House in 1961, the middle cranial fossa approach (MCFA) has been performed to access the internal auditory canal (IAC) while trying to preserve the rest of the auditory function [1]. In addition to intracanalicular vestibular schwannomas (VSs) close to the fundus, numerous surgical outcomes of the MCFA in facial nerve palsy, superior semicircular canal (SSC) dehiscence syndrome, and tegmen dehiscence have been reported [2,3,4,5,6]. Nonetheless, the MCFA is still considered an intricate approach because anatomical landmarks, such as the facial nerve, cochlea, and semicircular canal, are highly vulnerable and difficult to identify. Several senior experts have presented their strategy to find the landmarks in a safe and intelligent way [1,7,8,9].

Although the classical MCFA is in charge of an exclusive indication for accessing intracanalicular VS close to the fundus, it requires a large incision that causes scarring, temporalis muscle atrophy, and excessive temporal lobe retraction [10]. To date, several authors have reported the modified MCFA technique for SSC dehiscence repair and tegmen repair for cerebrospinal fluid (CSF) leakage to redeem such drawbacks [11,12,13,14]. Moreover, the advancement of adequate apparatuses, such as endoscopes and navigation systems, contributed to the development of minimally invasive techniques [15,16]. In this study, the authors aimed to expand the bounds of the minimally invasive technique that endoscopic-assisted keyhole MCFA (KMCFA) allows and re-highlight this technique in small intracanalicular VSs. To the best of our knowledge, this is the first trial for VS treatment using the KMCFA.

## 2. Materials and Methods

### 2.1. Cadaveric Demonstration

For comparison with the classical MCFA, the authors determined the capable exposure and potential morbidity via the endoscopic-assisted KMCFA. The authors used the endoscopic-assisted KMCFA to reach the IAC and expose the 8th cranial nerve in four fresh adult cadaveric whole heads (eight sides). The anatomical structures were explored, and the distances between zygomatic root and ossicles and ossicles and geniculate ganglion were measured to identify the IAC through a small incision using a surgical microscope and endoscope for magnification. After the heads were placed turned contralaterally and the ipsilateral external auditory canal was exposed, a 3-cm horizontal incision was made at the level of the zygoma (Figure 1A). The incision’s midpoint was perpendicular to the extension of the tragus, i.e., the incision was 1.5-cm long back and forth around the tragus. Subcutaneous tissue, temporalis muscle, and periosteum were dissected to expose the zygomatic root and temporal bone (Figure 1B). Round keyhole craniectomy (2 cm in diameter) was performed centered on the midpoint of the skin incision by drilling under the squamosal suture line. Once the temporal lobe was exposed, it was gently dissected from the middle fossa floor. The tegmen tympani was opened to identify the malleus head as a landmark that Catalano et al. had described (Figure 1C) [17,18]. Catalano et al. reported that the distance between the zygomatic root and malleus head was 18 mm [17]. The IAC vertical crest was placed on the straight line connecting the zygomatic root and malleus head. The geniculate ganglion was approximately 6.5 mm away, at approximately 23 degrees anterior to the malleus head (Figure 2) [18]. We measured the anatomic space in our dissections. After the geniculate ganglion was identified, the labyrinthine facial nerve and fundus of the IAC could be successively traced. An endoscope could be used to reach the porus of the IAC and secure the view (Figure 1D). Under endoscopic guidance, the porus of the IAC was skeletonized close to the petrous ridge (Figure 1E).

### 2.2. Clinical Cases

The authors performed retrospective medical chart review of the patients who received the surgery. This study was approved by the Institutional Review Board of Severance Hospital (4-2021-1554) and conducted under the principles outlined in the Declaration of Helsinki. The IRB committee approved a waiver of informed consent from all patients.

Between March 2014 and February 2021, five and eight patients with VS underwent endoscopic-assisted KMCFA and classical MCFA surgery by a senior surgeon (ISM). In the KMCFA, the incision was made in the same way as in the cadaveric study (Figure 3A). The resected calvarium was stored for the later tegmen repair. The temporal lobe dura was elevated from the middle fossa floor to the superior petrosal sinus from posterior to anterior. The middle meningeal artery (MMA) was the anterior border of dissection and was not sectioned. After the area of an expected greater superficial petrosal nerve (GSPN) and arcuate eminence (AE) was exposed, the Yasargil self-retaining retractor was placed medial to the petrous ridge. The middle fossa floor, 1.5- to 2-cm medial to the outer cortex of the temporal squamosa, was drilled with a micro-diamond bur (Medtronic, Minneapolis, MN, USA) to find the ossicles without damage (Figure 3B). The malleus head, incus, geniculate ganglion, and IAC fundus were identified consecutively under the guidance of an endoscope and a facial nerve monitoring device (Medtronic, Minneapolis, MN, USA). The IAC was meticulously skeletonized with micro drills (Medtronic, Minneapolis, MN, USA) from the fundus of the IAC to the posterior fossa in the shape of a funnel. For the KMCFA, this type of equipment is essential for manipulating structures in the narrow corridor. The IAC dura was exposed circumferentially and uncovered from the acoustic-facial bundle (Figure 3C). The exposed VS was totally removed. The cochlear and facial nerves were preserved (Figure 3D). The IAC was sealed with the temporalis muscle and fascia. The tegmen opening was covered and sealed with resected calvarium and fibrin glue (Figure 4A). Lastly, the cranium opening was covered with a dime-sized titanium plate and screws (Figure 4B). In the classical MCFA, the posterior-based skin incision started behind the hairline, was 6 × 6 cm wide, and was square-shaped, as depicted in Figure 5 with the blue line. Craniotomy was centered on the zygomatic root and performed using a cutting bur with a 4.5 × 4.5 cm wide bone flap. Dura was elevated along the middle fossa floor to expose the MMA, GSPN, AE, and petrous ridge at a glance. After identifying the structures above, the following steps were similar to those in the KMCFA; however, the surgery was completed without the endoscope. During the surgery, there were no additional CSF drainage procedures, such as lumbar puncture or ventriculoperitoneal shunt.

### 2.3. Outcome Evaluation

Preoperative and immediate postoperative magnetic resonance images were taken to evaluate the extent of tumor removal. Operation time, length of hospital stay, and postoperative complications were assessed. Pure tone audiometry (PTA) was performed for audiological evaluation, and the average PTA threshold across the four frequencies (500, 1000, 2000, and 4000 Hz) was calculated. Word recognition score (%) was measured at the most comfortable level. Fifty monosyllabic words from Hahm’s list were used, which comprised phonetically balanced Korean standard words [19]. One month before and after the surgery, the audiometric class was depicted based on the Committee on Hearing and Equilibrium of the American Academy of Otolaryngology–Head and Neck Surgery guidelines (2012) [20].

### 2.4. Statistical Analyses

All statistical analyses were performed using SPSS software version 21 for Windows (IBM Corp., Armonk, NY, USA). Mean and standard deviation were used for descriptive statistics. The two- sample *t*-test or the Mann–Whitney U test was used to compare the KMCFA and classical MCFA. Furthermore, the outcomes were tested using Fisher’s exact test.

## 3. Results

### 3.1. Cadaveric Demonstration

There was no intracranial disease seen in any of the four cadavers. The endoscopic-assisted KMCFA successfully exposed the intracanalicular structures without violating the ossicles, cochlea, and SSC. The distance between the zygomatic root and malleus head was 17.8 ± 1.44 mm. The distance between the malleus head and geniculate ganglion was 6.49 ± 0.62 mm. With the keyhole craniectomy corridor, although the surgical space to manipulate the micro drill, suction irrigator, and other surgical devices was subtly restricted to skeletonize the IAC, it was possible to explore the porus of the IAC with the microscopic approach alone. Nonetheless, the endoscopic visualization could provide a broad and brightened surgical perspective.

### 3.2. Clinical Cases

Five patients underwent the endoscopic-assisted KMCFA and eight underwent the classical MCFA (Table 1). The mean age was 57.57 ± 6.19 and 45.1 ± 5.32 years, and the preoperative mean hearing threshold was 40 ± 10.0 and 28 ± 7.4 dB HL in the KMCFA and classical MCFA groups, respectively (Table 2). The tumors were entirely confined to the IAC and were close to the fundus (Koos grade I) in all patients. Gross total resection was achieved in all KMCFA and in 62.5% (5/8) of classical MCFA cases. None of the patients experienced postoperative facial weakness, CSF leakage, headache, or seizure in the KMCFA group (0%, 0/5). Two patients (25%, 2/8) suffered postoperative complications in the classical MCFA group (one patient had postoperative facial palsy and severe headache and another patient had postoperative CSF leakage). Middle ear effusion was not identified, and the scar was not visible 1 month after the surgery in all patients. The mean operation time and the average length of hospital stay in the KMCFA group were shorter than those in the classical MCFA group (Table 2, *p* = 0.06 and *p* = 0.005). The hearing outcomes were similar to those of the classical MCFA group (perceived as an acceptable range in two cases, Appendix A).

## 4. Discussion

The MCFA, undoubtedly, could be the primary choice for neurotologists to consider when preserving hearing in cases of intracanalicular VS close to the fundus [21,22,23]. In this study, with endoscopic-assisted KMCFA, the authors drastically reduced the incision and craniotomy size and enhanced visualization (Figure 5). This breakthrough can minimize the sequelae from the classical MCFA, as well as counter the surgeon’s disfavor for the classical MCFA.

The MCFA has been frequently compared to the retrosigmoid approach in terms of hearing preservation. However, it differs from the retrosigmoid approach in indication, depending on the tumor location, because it can access the entire IAC. Radiosurgery has advanced in recent years and its hearing preservation rate has been reported to be 15–85% by miscellaneous assessments during various follow-up periods [24,25,26,27,28]. Therefore, intracanalicular VS is often treated with radiosurgery or even managed with watchful waiting. To some extent, the MCFA seems to be a “lost art” in VS treatment [29]. There could be several reasons for this. First, the tumor rarely grows, and its symptoms could be stable, especially in intracanalicular VS [30,31]. Second, the treatment cost could be an issue based on the medical circumstances in each country [32]. Moreover, the patients’ fear of sequelae from the surgery or the open cranium surgery itself, as well as the surgeons’ preference and skill, on which the results of facial nerve and hearing preservation depend, cannot be ignored [33,34]. If the surgeon and patient agree to the MCFA over the conditions outlined above, the advantages of this approach would be definite. Moreover, the present study might further reinforce the merits of the classical MCFA.

Using the KMCFA, first, postoperative headache and seizure development were minimized by reducing the range applied for temporal lobe retraction, which was assisted by a more definite and straightforward landmark and endoscope application. The authors could quickly find the major landmarks using the tragus, malleus, and geniculate ganglion. The endoscope could also secure the porus of the IAC with spacious aspects. This made it possible to access the IAC via a narrower but safer and more definite corridor, which enabled the delicate preservation of facial nerve and hearing functions (Figure 6). Traditionally, the MCFA has been considered a contraindication in elderly individuals over the age of 60 years because of the vulnerable dura [6]. However, Kohlberg et al. recently reported that there was no significant difference in postoperative recovery, outcome, and complications between the elderly and the young patient groups after the MCFA [34]. In this aspect, the KMCFA can expand its indications in elderly individuals by reducing dural manipulation. We also experienced no different outcomes in a 77-year-old patient using our modified technique. Second, a small craniotomy reduced the surgeon’s surgical time and labor. For large openings, surgeons need to minimize bone loss and make a well-designed bone flap for restoration. However, we do not need to design or preserve good bone flaps for small openings, and the drilling area is significantly reduced. Moreover, as Catalano et al. and Lee et al. have reported, the distance from malleus head to zygomatic root was highly accordant in our study [17,18]. Even paradoxically, using surface landmarks, small openings alleviate surgeons’ confusion about craniotomy locations and force them to make a quick decision. Although the statistical significance was not reached, the operation time was shorter than that of the classical MCFA. Lastly, cosmesis is one of the most significant advantages of this minimally invasive technique that cannot be overlooked.

The limitations of the endoscopic KMCFA are not too different from those of the classical MCFA. Meticulous bleeding control is required to obtain the optimal corridor when elevating the dura from the skull base. If the tumor originates from the inferior vestibular nerve, the operation will be difficult. Furthermore, because the working space for controlling the micro drill is relatively narrow, great care should be taken not to injure the susceptible structures, especially the anteriorly deviated facial nerve due to the tumor. In our cases, although the tumors were removed without any major postoperative complications in the KMCFA, the hearing preservation rate was still not promising. The authors speculated that this could be because of microscopic damage to the small vessels supplying the cochlear nerve during tumor dissection. Although the hearing outcomes may vary depending on the institution, the results may benefit surgeons who are already accustomed to the MCFA [33]. Further, an endoscope has only one lens, and depth perception might be restricted compared to that with a microscope with binocular vision. This inherent limitation of endoscopes should be considered in this study. Therefore, the KMCFA was successful when both optical devices were properly used. Lastly, KMCFA could be challenging for surgeons unfamiliar with the obscure anatomy of the skull base. In the study, the senior surgeon who performed KMCFA was experienced not only at the classic MCFA but also at the minimally invasive technique for other indications, such as facial nerve schwannoma and SSC dehiscence. Thus, for successful KMCFA, considerable learning experience is a prerequisite.

To the best of our knowledge, this is the first trial for VS treatment using the KMCFA. The anatomical knowledge attained from the cadaveric demonstrations and intraoperative findings suggests that the KMCFA may help remove intracanalicular VSs in a safe, efficient, and cosmetic way.

## Figures and Tables

**Figure 1 jcm-11-02324-f001:**
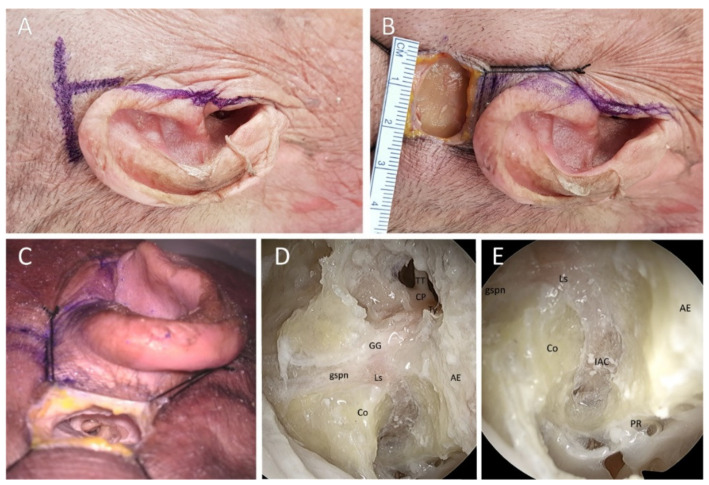
Cadaveric specimen, right ear. (**A**): 3-cm linear horizontal incision is made at the level of the pinna. The incision is 1.5-cm-long back and forth around the tragus, which is in line with the zygomatic root. (**B**): After skin retraction, the entrance of the keyhole middle cranial fossa approach is approximately 2 cm in diameter. The temporalis muscle and periosteum are resected for convenience during the dissection. (**C**): The tegmen tympani is opened, and the ossicles are identified. The zygomatic root exists between the malleus head and the line extending the tragus. (**D**): Endoscopic view shows the tensor tympani, malleus (not shown in this picture), facial nerve (labyrinthine segment, geniculate ganglion, and tympanic segment), and arcuate eminence under the bright sight. (**E**): A 30-degree endoscope helps identify the IAC. The whole length of the facial nerve in the IAC is exposed close to the petrous ridge. AE: arcuate eminence, Co: cochlea, CP: cochleariform process, GG: geniculate ganglion, gspn: greater superficial petrosal nerve, IAC: internal auditory canal, Ls: labyrinthine segment of the facial nerve, TT: tensor tympani muscle, Ty: tympanic segment of the facial nerve, PR: petrous ridge.

**Figure 2 jcm-11-02324-f002:**
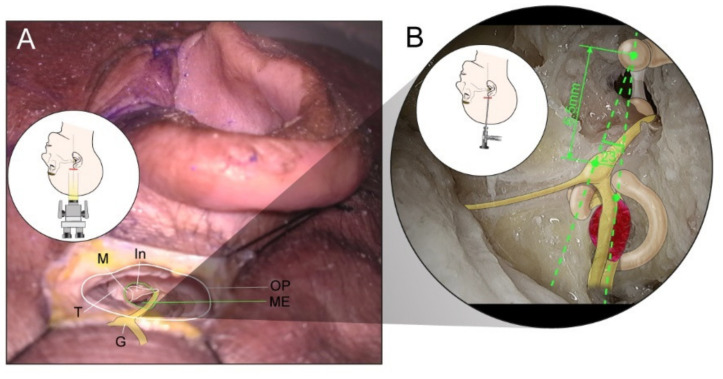
A cartoon is superimposed on the picture to compare the difference in visual field between the microscope and the endoscope. (**A**): Major landmarks in the middle ear cavity can be easily identified using a microscope, and there is almost no limit to the field of view. (**B**): The endoscope allows the visualization of the entire operation field, including the whole IAC from the fundus to the porus. The IAC vertical crest is placed on the straight line connecting the zygomatic root and the malleus head, and the geniculate ganglion is approximately 6.5 mm away, at about 23 degrees anterior to the malleus head. G: greater superficial petrosal nerve, In: incus, M: malleus, ME: middle ear cavity through middle fossa tegmen defect, OP: keyhole entrance for operation after craniectomy,T: tensor tympani, IAC: internal auditory canal.

**Figure 3 jcm-11-02324-f003:**
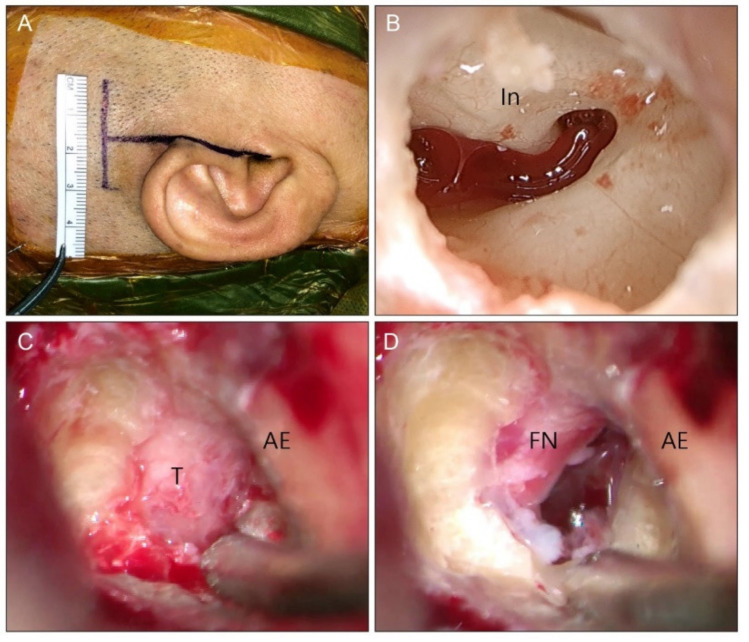
Clinical case, right ear. (**A**): A 3-cm linear horizontal incision is made similar to that in the cadaveric study. (**B**): Endoscopic view shows the incus and tympanic segment of facial nerve through the opened tegmen tympani. The location of the geniculate ganglion can be presumed. (**C**): The internal auditory canal has been skeletonized, and the dura is uncovered. The tumor is seen displacing the facial nerve anteriorly. (**D**): After gross total resection of the tumor, the facial nerve is preserved. AE: arcuate eminence, FN: facial nerve, In: incus, T: tumor.

**Figure 4 jcm-11-02324-f004:**
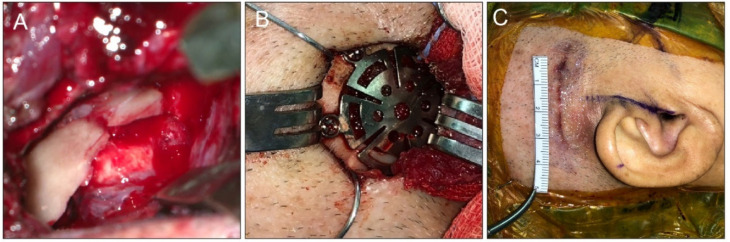
Clinical case, right ear. (**A**): The opened tegmen tympani is reconstructed with the cortical bone chip. Furthermore, the internal auditory canal is secured with temporalis muscle and fascia. (**B**): The craniectomy site is covered with a titanium plate and screws. (**C**): The final skin wound is closed using a skin adhesive bond. Since the wound is on the scalp, it will be covered when the hair grows.

**Figure 5 jcm-11-02324-f005:**
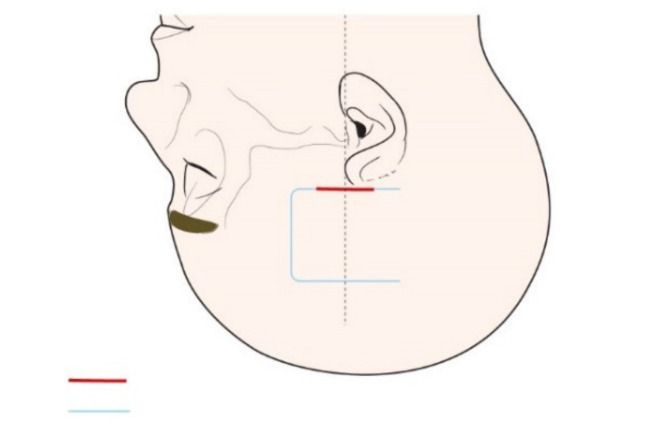
Comparison of the openings of the classical middle cranial fossa approach (MCFA, blue line) and the keyhole MCFA (red). The keyhole MCFA drastically reduces the incision and craniotomy size.

**Figure 6 jcm-11-02324-f006:**
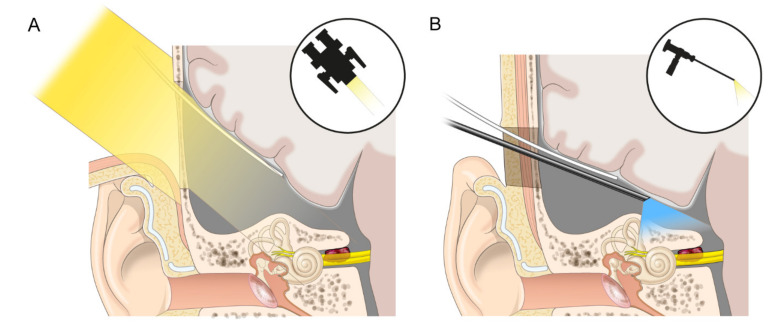
Comparison of the field between classical middle cranial fossa approach (MCFA) and keyhole MCFA. (**A**): The classical MCFA needs to retract the brain sufficiently to secure all the landmarks and the tumor, albeit the light from the microscope is attenuated. (**B**): In the keyhole MCFA, an entire field of view can be secured by placing the endoscope close to the lesion despite a small opening and slight retraction.

**Table 1 jcm-11-02324-t001:** Summary of clinical cases.

Surgery	Patient	Sex	Age (Years)	Site	Tumor Size (mm)	Tumor Growth Rate (mm/year)	Intractable Vertigo	Preoperative Hearing Threshold (dB HL)	Preoperative Word Recognition Score (%)	Hearing Deterioration Rate (dB/6 Months)	Surgical Outcome	Postoperative Complications	Operation Time (Minutes)	Hospital Stay (Days)
KMCFA	1	Male	40	Left	8 × 5	Not available	No	68	58	29	Total	None	312	6
KMCFA	2	Female	60	Left	11 × 8	2	Yes	26	90	2	Total	None	345	7
KMCFA	3	Female	77	Left	7 × 3	2	No	61	26	5	Total	None	263	10
KMCFA	4	Female	49	Right	4 × 5	<1	Yes	8	96	None	Total	None	501	7
KMCFA	5	Male	59	Left	9 × 5	Not available	No	56	70	2	Total	None	320	7
MCFA	6	Male	49	Left	7 × 4	1	Yes	10	94	None	Total	Facial palsy, headache	427	9
MCFA	7	Male	59	Right	13 × 8	Not available	Yes	24	90	Not available	Subtotal	None	637	9
MCFA	8	Female	37	Left	8 × 5	Not available	Yes	10	Not available	Not available	Subtotal	None	681	9
MCFA	9	Male	44	Left	6 × 4	<1	No	60	52	3	Subtotal	None	555	11
MCFA	10	Female	51	Left	6 × 2	Not available	No	33	88	Not available	Total	None	430	15
MCFA	11	Female	38	Left	5 × 4	No growth	No	58	70	Not available	Total	CSF leakage	405	12
MCFA	12	Female	66	Left	3 × 2	Not available	No	8	98	Not available	Total	None	460	12
MCFA	13	Male	17	Right	5 × 3	No growth	No	21	98	Not available	Total	None	300	11

MCFA: classic middle cranial fossa approach, KMCFA: keyhole middle cranial fossa approach.

**Table 2 jcm-11-02324-t002:** Comparison of patients who underwent VS removal via the KMCFA (*n* = 5) and classical MCFA (*n* = 8).

	KMCFA	Classical MCFA	*p*-Value
Sex (male:female)	2:3	4:4	
Site (left:right)	4:1	6:2	
Age (years)	57 ± 6.19	45.1 ± 5.32	0.18
Tumor size (mm)	8 ± 1.00	6.8 ± 1.04	0.45
Tumor growth rate ^†^ (mm/6 months)	2	0.3	
Intractable vertigo rate (%)	40 (2/5)	37.5 (3/8)	
Preoperative hearing threshold (dB HL)	40 ± 10.0	28 ± 7.4	0.35
Surgical outcome (rate of total removal, %)	100 (5/5)	62.5 (5/8)	0.23
Major complication ^‡^ rate (%)	0 (0/5)	25 (2/8)	0.49
Surgical time (minutes)	348.2 ± 40.45	486.9 ± 45.13	0.06
Hospital stay ^§^ (days)	7.4 (6.5–8.5)	11 (9–12)	0.0054 **

^†^ Only three KMCFA and four classical MCFA patients’ data were available. ^‡^ Major complications include facial palsy, cerebrospinal fluid leakage, postoperative bleeding, headache, and seizure. ^§^ Mann–Whitney U test. ** *p*-value less than 0.01. KMCFA: keyhole middle cranial fossa approach, MCFA: middle cranial fossa approach; VS: vestibular schwannoma.

## Data Availability

The data are available from the corresponding author upon reasonable request.

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
