# Peer review of "Endoscopic-Assisted Keyhole Middle Cranial Fossa Approach for Small Vestibular Schwannomas"

_jcm, 2022, doi:10.3390/jcm11092324_

Round 1

Reviewer 1 Report

This manuscript addresses the efficacy of keyhole middle cranial fossa approach (KMCFA) with endoscopic assistance for intracanalicular vestibular schwannomas. The authors demonstrate the anatomical structure and special features such as narrow corridors and landmarks in this approach using cadaveric heads. The methodology of KMCFA is clearly presented. KMCFA applied clinically was not inferior to the classical middle cranial fossa approach, and even had an advantage of minimal invasiveness. Although I do not wholly agree with the authors’ idea of applying this approach for vestibular schwannomas, not for facial schwannomas, this paper is worthy of eventual publication.  

The authors recognized that surgical time had reduced thanks to KMCFA. According to Table 2, it does not seem to be significantly different with p-value of 0.06. They should also clarify the threshold of significance such as “p < 0.05”. Moreover, the caption of the double-asterisk in Table 2 is needed.

Author Response

This manuscript addresses the efficacy of keyhole middle cranial fossa approach (KMCFA) with endoscopic assistance for intracanalicular vestibular schwannomas. The authors demonstrate the anatomical structure and special features such as narrow corridors and landmarks in this approach using cadaveric heads. The methodology of KMCFA is clearly presented. KMCFA applied clinically was not inferior to the classical middle cranial fossa approach, and even had an advantage of minimal invasiveness. Although I do not wholly agree with the authors’ idea of applying this approach for vestibular schwannomas, not for facial schwannomas, this paper is worthy of eventual publication.  

>> I would like to thank the reviewer for the thorough and helpful review of the manuscript. We have made changes to address all of the points that were raised. Thank you again.

The authors recognized that surgical time had reduced thanks to KMCFA. According to Table 2, it does not seem to be significantly different with p-value of 0.06. They should also clarify the threshold of significance such as “p < 0.05”. Moreover, the caption of the double-asterisk in Table 2 is needed.

>> The operation time seems different between the two groups. However, as it is not statistically significant, the conclusion drawn was a bit of an overstatement. We have added the p-value in the Results section and corrected the description in the Discussion section as advised by you. Further, an explanation for the meaning of the double-asterisk was added to Table 2. 

Reviewer 2 Report

The authors report a case series of patients affected by intracanalicular vestibular schwannoma using an endscopic key-hole middle cranial fossa approach, highlighting its advantages and limitations compared to the traditional one. The manuscript appears appropriately structured, and the association with the cadaveric study makes it relevant from a scientific point of view.
Just a few observations: 1) Certainly the key-hole endoscopic approach at the level of the middle cranial fossa requires a considerable learning curve. This issue should be explored and mentioned in the paper, although surgeries were performed by a senior surgeon. 2) Specify the "safe" role of radiosurgery in hearing preservation, also in terms of percentage and case studies reported in literature.
3) Quantify the rate of post-operative complications using this endoscopic approach (headache, epilepsy, CSF fistulas)

Author Response

The authors report a case series of patients affected by intracanalicular vestibular schwannoma using an endscopic key-hole middle cranial fossa approach, highlighting its advantages and limitations compared to the traditional one. The manuscript appears appropriately structured, and the association with the cadaveric study makes it relevant from a scientific point of view.

>> I would like to thank the Reviewer for the thorough and helpful review of the manuscript. Authors have made changes to address all of the points that were raised. Thank you again.

Just a few observations:

1) Certainly the key-hole endoscopic approach at the level of the middle cranial fossa requires a considerable learning curve. This issue should be explored and mentioned in the paper, although surgeries were performed by a senior surgeon.

>> MCFA itself is a highly difficult operation, which can be performed only by an experienced surgeon. Therefore, KMCFA involves a long and complicated learning curve, as you have mentioned. As advised by you, the authors have described these points in the discussion section.

2) Specify the "safe" role of radiosurgery in hearing preservation, also in terms of percentage and case studies reported in literature.
>> Thank you for your comment. The authors totally agree that radiosurgery has its own merits in terms of hearing outcomes, especially for small intracanalicular tumors distant from the cochlea. In this study, the authors did not assess the superiority of microsurgery against radiosurgery but hoped to introduce the modified and advanced surgical technique as one of the options. Accordingly, the information on brief hearing preservation rate, noted from various relevant pieces of literature, was added to the manuscript, while keeping in mind the scope and not detracting from the contextual issue, in the discussion section (lines 204-205).

3) Quantify the rate of post-operative complications using this endoscopic approach (headache, epilepsy, CSF fistulas)

>> AWe have further described the rate of complications in the results section (3.2. clinical cases). Detailed information has also been provided in Table 1.